# Overview of the Available Knowledge for the Data Model Definition of a Building Renovation Passport for Non-Residential Buildings: The ALDREN Project Experience

**Marta Maria Sesana [1],*** **, Mathieu Rivallain [2] and Graziano Salvalai [3]**

1   Polo Territoriale di Lecco, Politecnico di Milano, 23900 Lecco, Italy
2   Centre Scientifique et Technique du Bâtiment, CEDEX 16, 75782 Paris, France; mathieu.rivallain@cstb.fr
3   Department of Architecture Built environment and Construction engineering, Politecnico di Milano, 20133 Milan, Italy; graziano.salvalai@polimi.it
*   Correspondence: marta.sesana@polimi.it

**Abstract:** According to its strategic long-term vision, Europe wants to be a climate-neutral economy by 2050. Buildings play a crucial role in this vision, and they represent a sector with low-cost opportunities for high-level $CO_2$ reduction. The challenge the renovation of the existing building stock, which must be increased to 3%/year, more than double compared to the current 1.2%/year. In this context, the ALliance for Deep RENovation (ALDREN) project has the goal of encouraging property owners to undertake renovation of existing buildings using a clear, robust, and comparable method. This paper aims to present the ALDREN approach and the ALDREN Building Renovation Passport (BRP), giving an overview of the connections and data links to other existing databases and certification schemes. To understand the data value potential of buildings, one requires reliable and trustworthy information. The Building Renovation Passport, introduced by the recent Energy Performance Building Directive (EPBD) recast 844/2018/EU, aims to provide this information. This paper presents the experience of the ALDREN BRP for non-residential buildings as well as the development procedure for its data model and the potential that this tool could have for the construction market. The ALDREN BRP has been structured into two main parts—BuildLog and RenoMap—with a common language that facilitates communication on the one hand and, on the other, the setting of renovation targets based on lifetime, operation, and user needs.

**Keywords:** energy performance; building value; energy saving; building renovation passport; non-residential buildings

## 1. Introduction

Over two years ago, a consortium of partners across numerous EU countries came together to complete a proposal to Horizon 2020 research program EE-11-2016-2017, titled ALliance for Deep RENovation, in buildings implementing the European Common Voluntary Certification Scheme, as a back-bone for the whole deep renovation process. The ALliance for Deep RENovation (ALDREN) goals consist of increasing renovation rates by overcoming market barriers and preparing the ground for investment. The overarching ambition of the ALDREN initiative is to consolidate, promote, and implement an extended harmonized procedure and a set of instruments to support deep building renovation operations by tackling organizational, financial, and technical issues. There is a paradox between the need for deep renovation (encouraging building energy retrofitting in a step-by-step

approach) and the scarcity of financial resources, the risk being that the potential for energy savings and the ensuring of an intermediate long-term energy performance level is missed.

The European Commission has called for a climate-neutral Europe by 2050, identifying the building sector as a valuable and feasible keystone in reaching the goal. To meet this target, the energy efficiency of existing buildings must be increased by at least 75% by 2030 [1]. In this context, the ALDREN project aims to motivate the construction sector value chain stakeholders to undertake deep renovation projects on their properties. Non-residential buildings, considering the percentage of floor area distributed across Europe (around 25%), could be the key solution to boost the energy renovation strongly requested by the European Commission and its latest building regulations.

The ALDREN project will provide approaches for step-by-step renovation and rules to take into account the valorization of energy efficiency, health, and well-being indicators in commercial real estate (i.e., the asset value of the building), in line with the development of a Building Renovation Passport (BRP) for non-residential buildings (especially offices and hotels) as a complementary tool to the Energy Performance Certificate (EPC).

The ALDREN BRP will include a tangible long-term vision for national renovation strategies with a link to finance instruments and minimum renovation targets, offering a secure pathway that avoids the lock-in effect towards $CO_2$ reduction. Long-term planning with intermediate targets would be very helpful for owners and stakeholders, increasing the confidence required to invest in renovation. The BRP will also be used for data collection regarding building characteristics, energy efficiency, and comfort levels, overcoming the lack of information that typically affects our buildings. The availability of building information regarding past and current interventions and retrofitting operations are fundamental to the reduction of the costs of future intervention.

The ALDREN approach, and the outcomes presented in this paper reflect the structure of the activities carried out during Task 2.6 of WP2, the technical work package of the ALDREN project composed of six tasks: four vertical (sectorial) tasks (T2.2 Energy rating and target, T2.3 Addressing the gap between calculated and actual EP; T2.4 Addressing health and well-being, and T2.5 Linking EVCS to the financial valuation), harmonized by a common language defined in the first horizontal task (T2.1) and integrated by an overall approach—the second horizontal one (T2.6)—into a Building Renovation Passport (BRP).

Task 2.6 is then sub-structured in three respective subtasks:

Task 2.6.1　Overview of the available knowledge and information;
Task 2.6.2　Rendering and structuring of all the collected data and knowledge;
Task 2.6.3　Building Passport (BP) form definition.

This paper presents the activities related to the analysis performed on the available knowledge of non-residential buildings, and a critical review of the results that lead to the definition of the data model structure for the ALDREN BRP. The overview of existing databases and similar ongoing initiatives regarding BRP is a very important task to compare the ALDREN BRP with comparable data, recognized standards, and compatibility with existing databases at national and European levels. This work aims to present the ALDREN approach and the ALDREN BRP, focusing on the connections and data links with other existing databases, certification schemes, and the potential that this service could have for the market.

The paper is structured as follows: Section 1 introduces first the ALDREN project in a nutshell and then focuses on the ALDREN concept and the presentation in more detail of Task 2.6 goals, the core of the paper; Section 2 sets the scene for the main topic data in the construction sector, providing an introduction to definitions, terminology, and criteria for the database analysis performed in detail in Section 3. The overview of the available knowledge then focuses on the currently ongoing European BRP initiatives and is presented in Section 4. Section 5, starting from the lessons learned from previous sections, presents the Data Model (DM) structure of the ALDREN BRP. Finally, conclusions and recommendations for next steps are made in Section 7.

## 2. Data terminology and Criteria for the Evaluation of Existing Buildings Database in Europe

As the actual value of data continues to grow, the limits of its prospective value are pushed every day. In 2017, the Economist [2] published a widely referenced article claiming that oil had been replaced as the world's most valuable resource by data—specifically, data captured from users by tech companies. Similarly, in the construction industry, data refers to the huge quantities of information that have been stored in the past and that continue to be acquired today. Data can come from everywhere, for example, people, sensors, machines, and any other data-generating devices, and they can be replicated indefinitely and moved around the world speedily, cheaply, and through the net.

Building construction data already exist in all the plans and records of anything that has ever been built. It is also constantly increasing with additional input from sources as diverse as on-site workers, cranes, earth movers, material supply chains, and even buildings themselves; however, ultimately there is no standardized way to archive, use, or share data, causing a general lack of understanding, transparency, and uniform methods when it comes to the overall process in the real estate market and consequently causing lack of confidence in owners or policy makers in relation, for example, to the effectiveness of technological strategies, the return of investments, or energy consumption reductions.

Data need to be curated, coddled, stored, and processed so they may be transformed into information and further refined into knowledge.

Considering the building sector, evaluation is highly dependent on good, reliable, and accessible data. This is equally true for the evaluation of energy programs and policies. Evaluators typically will try to leverage as much available data, databases, and studies as possible to support their evaluations.

This section presents some key database concepts and their corresponding definitions, considering the basic structure of the Data, Information, Knowledge, and Wisdom (DIKW) pyramid. Like other hierarchy models, the DIKW pyramid has rigidly set building blocks—data comes first, information is next, then knowledge follows, and finally wisdom is on the top. Each step up the pyramid answers questions about the initial data and adds value to them.

The selection and the description of the terms have been studied and individuated from the literature for this topic [3,4] and summarized in Table 1. This selection was performed with two objectives:

(i)     Providing a jumping off point for those interested in learning more about databases and their design and management;
(ii)    Focusing on the concept that affected the definition of the ALDREN BRP data model.

Nowadays there are multiple databases related to building characteristics and stocks. The building's data are collected by different institutions (i.e., statistics offices, energy agencies, consultancy companies, research organization, others) mainly on the member states (MS) level; its quality, availability, and completeness vary significantly between the different countries [1].

An overview of a selection of existing databases has been conducted with the aim to individuate their main characteristics and to understand which of them can be used, linked, and connected to support the ALDREN methodology for the BRP definition and the assessment of energy interventions in existing non-residential buildings. The main collectors of these data, considered relevant for ALDREN approach at European level, have been investigated and analyzed referring to the criteria defined in Table 1.

**Table 1.** Main criteria and parameters used for database analysis.

| Code | Criteria | Description |
|------|----------|-------------|
| A | Organization of the data | Data collected can be stored and organized in different ways. Typical choices for data organization are the following.<br>• Relational database: in this database the raw data is organized into sets of tables, and into relations between them:<br>• NoSQL database: NoSQL is an umbrella term, which encompasses several different technologies that are not relational in nature. This lack of relational structure results in unstructured or semi-structured data in storage.<br>• No specific organization: data are stored without a specific machine-readable organization criterion. |
| B | Organization of the database | Databases for data collection can be stored in different ways. Focusing on the Energy Performance Certificate (EPC) register, the most common database organization can be as listed following:<br>• National database: a national building database exists and all the EPCs are collected in this unique database (DB);<br>• Regional database: a regional building database exists and all the EPCs are collected in this unique DB;<br>• Multiple databases: multiple databases exist (both at a national and regional level) for each specific EPC;<br>• No collection in database: EPCs are not collected in a database. |
| C | Organization responsible for the database | To get access to the information in the database, it is fundamental to know the organization responsible for the data. This is mainly connected to the organization of the database (see previous point).<br>• National authority: one or more national public authorities collect data and maintain the database.<br>• Regional authority: one or more regional public authorities collect data and maintain the database.<br>• Private company: one or more private companies collect data and maintain the database.<br>• Semi-public company: one or more semi-public companies collect data and maintain the database. |
| D | Data collection | Data collection methods can give important information about data quality.<br>• Automatic filling: data are automatically reported in the database by a certification/calculation tool that can be managed by consultants/experts.<br>• Centralized reporting: a central secretary is reporting the data from the collection to the database.<br>• Simple Reporting: data are reported in the database by consultants/experts. |
| E | Data quality | If a high number of independent checks exist, the quality of the data is supposed to be verified in one of the following modes.<br>• Statistical check: generation of statistical information and cleaning of data and identification of out-of-range data after input.<br>• Probability check on entry: e.g., acceptable value range for different parameters.<br>• Crucial data check: data from a certificate is rejected if crucial data are missing.<br>• Syntactical check: e.g., no text in numerical fields.<br>• Other. |
| F | Data query | Possibility to perform queries on the data.<br>• Database supports query: all fields of the database can be queried.<br>• Database partially supports query: only a limited number of fields of the database can be queried.<br>• Database does not support query: database does not allow third party queries. |
| G | Connection with other databases and other uses | The database is connected to the information source, e.g., spatial data infrastructure: Yes/No |

Following a brief presentation to set the scene on the database's overview, information is presented in more detail in the following dedicated subsections: (Section 3.1) Eurostat, (Section 3.2) building stock observatory (BSO), (Section 3.3) energy performance certificate (EPC) registers, (Section 3.4) building energy performance data platforms outcomes of European projects.

Eurostat [5] is an official directorate-general of the European Commission (EC) with the main responsibility to provide statistical information to the institutions of the European Union (EU) and to promote the harmonization of statistical methods across its MS through the European Statistics System (ESS). Eurostat statistics are open data available on its website; they are hierarchically ordered in a navigation tree with tables distinguished from multi-dimensional datasets from which the statistics are extracted via an interactive tool.

Building stock observatory (BSO) it the official and centralized database on the building's stock for the European area. The Observatory was developed for the European Commission by the Buildings Performance Institute Europe (BPIE) in collaboration with ECN, Ecofys, Enerdata, and SEVEn, as well as national project partners [6]. BSO contains a database, a data mapper, and factsheets. BPIE began collecting facts and figures on the European building stock in 2010 in the context of preparing its major study of Europe's buildings [7].

In 2012, the information gathered was made available in the data hub portal (by BPIE) including technical data on building performance. The platform, no longer available, offered country statistics on building's monitoring of the EU directive implementation [8]. The data hubs were used as a basis to design the BSO; currently, the BSO is managed by the RICS organization (Royal Institution of Chartered Surveyors).

The energy performance certificate (EPC) was established in 2002 with the EPBD [9] and its implementation process has been remarked in 2010 [10] to stimulate the creation of an independent register, often at local level. Databases collecting data from EPCs are the main source regarding the energy performance of the EU building stock. Unfortunately, some European countries do not have central databases, and where they do, they are not usually public access [11].

Finally, in Section 3.4, a selection of platforms related to buildings' energy performance data realized within European researches (mainly belonging to the 7th Framework Programme funded European Research and Technological Development FP7 and Horizon 2020 Framework Programme for Research and Innovation), are presented and analyzed considered as a valuable sources for both the overview of the available knowledge.

The following have been selected as the most related to the ALDREN project and further investigated: Section 3.4.1 EPISCOPE-TABULA; Section 3.4.2 REQUEST2ACTION; Section 3.4.3 EXCEED referring to the criteria of Table 1.

More comprehensive analyses on all the above listed databases have been presented in each respective subsection.

## 3. Overview on the Available Knowledge on European Databases

### 3.1. Eurostat

Eurostat is the agency within the European Union (EU) charged with providing statistical information for the continent and ensuring that member countries are using acceptable methods to track and report statistics within their borders. Eurostat consolidates, processes, and standardizes all statistical data from member countries so that they are cross-comparable conceptually and in terms of units of measurement. Eurostat analysis results are presented in Table 2.

**Table 2.** Eurostat analysis by criteria defined in Table 1.

| Code | Criteria | Description |
|------|----------|-------------|
| A | Organization of the data | Eurostat covers all areas of European society with over 4600 datasets containing more than 1.2 billion statistical data values. The data are available for consultancy by a navigation tree and all datasets are presented both as data and graphs. |
| B | Organization of the database | The Eurostat statistical work is structured into themes and sub-themes. The main statistical areas are nine: general and regional statistics; economy and finance; population/social conditions, industry, trade/services; agriculture/fisheries; international trade; transport; environment/energy; science, technology, and digital society. There are also some cross-cutting topics intercorrelated on sub-themes. The data are available in tables which are easy to export. |
| C | Organization responsible for the database | Eurostat is the statistical office of the EC born with the mission to provide high quality statistics for Europe. |
| D | Data collection | Eurostat data contains several indicators on the EU-28 and the Eurozone, the member states and their partners. The database of Eurostat contains always the latest version of the datasets meaning that there is no versioning on the data. Datasets are updated twice a day, at 11:00 and at 23:00, in case new data are available or because of structural change. It is possible to access the datasets through SDMX Web Services as well as Json and Unicode Web Services. |
| E | Data quality | Eurostat performs data validation by verifying whether data are in accordance with certain basic criteria that serve to assess the plausibility of the given data. In the target business process, validation rules are jointly designed and agreed upon at the level of each statistical domain's working group. The resulting validation rules are documented using common cross-domain standards, with clear validation responsibilities assigned to the different groups participating in the production process of European statistics. The European Statistical System (ESS) oversees the data validation workflow. |
| F | Data query | Eurostat database supports query: all fields of the database can be queried. |
| G | Connection with other databases and other uses | YES. The European Statistical System (ESS) is the partnership between the community statistical authority and other national authorities responsible in each Member State for the development, production, and dissemination of statistical data. |

### 3.2. The Building Stock Observatory (BSO) Database

The EU BSO is an EC initiative to monitor the energy performance of buildings across Europe. The purpose of the EU BSO is to (i) provide a snapshot of the energy performance of the EU building stock, by providing high-quality data from all MSs in a consistent and comparable manner and (ii) set a framework/methodology for the continuous monitoring of the building stock. The BSO primary

driver is to provide a clear understanding of the effectiveness of EU policy measures and mechanisms to enhance energy efficiency in Europe [12]. BSO analysis results are presented in Table 3.

**Table 3.** BSO analysis by criteria defined in Table 1.

| Code | Criteria | Description |
|---|---|---|
| A | Organization of the data | The BSO includes more than 250 indicators grouped in 10 thematic areas: i.e., building stock characteristics, building renovation, nearly Zero-Energy Buildings, energy consumption, building shell performance, technical building systems, certification, financing, and energy poverty and energy market. |
| B | Organization of the database | The BSO includes more than 250 indicators grouped in the following thematic areas: Topic 1.1, building stock—NZEB and new construction; Topic 1.2, energy needs; Topic 1.3, fuel supply mix; Topic 2, technical system; Topic 3, certification; Topic 4, financing; Topic 5, fuel poverty and social aspects; Topic 6, building code. A future topic is comfort. The BSO is designed to contain a wide variety of information in different formats (e.g., averages or totals), units (e.g., counts, shares or data-specific units) and aggregation levels (e.g., per building type, per energy source or per year) covering six main topic areas. Due to such details and the variety of the information, each data field has some properties to consider for data gap management unique to each indicator. |
| C | Organization responsible for the database | The first phase of the project, launched in November 2016, was developed by a consortium led by BPIE. The second phase is the continuation of the work on the EU BSO lead by RICS. |
| D | Data collection | The data collection for the BSO has been conducted mainly using data from Eurostat and from EU funded projects. Other data were collected by the consortium partners in collaboration with 20 national partners with the scope to gather specific data from each member state. In the original iteration of the EU BSO, numerous EU Projects were employed to populate approximately 4% of the entire database. However, those projects are finished and no longer collecting data, which consequently means that, the indicators—within the EU BSO that were originally populated through these projects—will not be updated with the latest data. |
| E | Data quality | The most used sources were the EU projects (around 40%) and official international or national statistics (around 39%), followed by expert assumptions and calculation provided by the national project partners and the consortium supported around 25% of the collected data. The EU BSO faces a serious amount of persistent data gaps. Some energy efficiency indicators present in the EU BSO have no data whatsoever. Currently, the EU BSO has 13% of its indicators populated with data. Of this 13%, 8.4% of the populated data are sourced regularly from high quality and reliable sources, i.e., Eurostat, Odyssee-Mure, and various National Statistical databases. |
| F | Data query | BSO supports queries; all fields of the database can be queried, but many datasets are still unavailable. Often it lacks years for some countries. All sources are provided in the public database and data visualization tools such as the factsheets. |
| G | Connection with other databases and other uses | Yes. Many data come from previous databases developed within EU project researches. Explicit references and links are available directly on the BSO browser during its navigation. |

## 3.3. EPC Registers

Table 4 provides the results of the analysis conducted as an overview on a selection of EPC-database developed at European level by the different MS referring also to the results of the EU project REQUEST2ACTION which focused on the following: Austria, Portugal, Netherlands, Slovakia, UK, and Italy with Regione Lombardia.

**Table 4.** EPC registers analysis by criteria defined in Table 1.

| Code | Criteria | Description |
|---|---|---|
| A | Organization of the data | The majority of the MS (18) collect EPC data in a central register managed by an official authority. Some MS do not have a central register and some MS appointed private company to populate and manage the system. The data filled in by the assessors into an accredited EPC tool are used directly to populate the database almost in all the MS. |
| B | Organization of the database | Databases are organized in different ways ranging from one country-wide database to several database organized at regional level (i.e., in Italy there are different EPC registers at regional level that provide data to the national based database (SIAPE)). |
| C | Organization responsible for the database | In most of the MS there is an official authority responsible for the database, either at a national level or at a regional level. In Austria, Slovenia, and Switzerland, a private company is in charge of running the database. |
| D | Data collection | In many cases, the data are entered directly by the assessor into the database or data comes directly from accredited certification tools. Then, usually all information collected during the certification process and audits in the buildings is stored in the database. |
| E | Data quality | The introduction of the EPC system in the first EPBD was not sufficiently supported by quality assurance requirements. In order to ensure high quality of energy performance certifications, an independent control system was introduced in the EPBD recast Art.18. There is not control before data entering into the database. EPC are controlled randomly after the completion. |
| F | Data query | The EPC register was born to facilitate the use of the data collected in databases for different purposes. Downloading or filtering select specific datasets is not always possible depending on the characteristics of each national/regional/local database. |
| G | Connection with other databases and other uses | The possibility to connect datasets to other databases depends on the data model and metadata characteristics of each national/regional/local. In some cases, for example, the regional register is linked to the national ones, there is still a lack of interoperability and standardization to implement. |

*3.4. Building Energy Performance Data Platforms Outcomes of EU Projects*

3.4.1. EPISCOPE-TABULA

The objective of the EPISCOPE project (presented in a nutshell in Table 5) was to make the energy refurbishment processes transparent and effective. The main outcome is a concerted and shared set of energy performance indicators able to ensure a high-quality renovation process in compliance with regulations, to track and steer the refurbishment processes in a cost-efficient way and to evaluate the impact of the actions monitoring the related energy savings. EPISCOPE-TABULA analysis results are presented in Table 6.

**Table 5.** EPISCOPE-TABULA projects in a nutshell.

| Title Project | Acronym | Duration | Database Name and Link |
|---|---|---|---|
| Energy Performance Indicator Tracking Schemes for the Continuous Optimization of Refurbishment Processes in European Housing Stocks | EPISCOPE | 04/2013 – 03/2016 | http://www.meteo.noa.gr/datamine/ |
| Typology Approach for Building Stock Energy Assessment | TABULA | 2009 – 2012 | http://webtool.building-typology.eu/#bm |

**Table 6.** EPISCOPE analysis by criteria defined in Table 1.

| Code | Criteria | Description |
|---|---|---|
| A | Organization of the data | TABULA project defines a common methodological structure for analyzing the residential building typologies. The typology has been classified according to their size, age, and energy-relevant parameters. |
| B | Organization of the database | An excel file was developed which consists of two tables with constant and variable input data. In parallel, a webtool was developed which enables an online calculation according to the TABULA method (TABULA WebTool). Each partner has its own division of building typology according to the needs of the national experts in the different application fields. The TABULA Data Structure is divided into: parameters for classification (country, region, construction year class, Building Size Class), reference area, calculation method building, boundary conditions, thermal envelope, U-values, consideration of thermal bridging, calculation method supply system, and delivered energy/fuel. |
| C | Organization responsible for the database | All TABULA partners are responsible for the consistent transformation between their national building typology and the common definition. |
| D | Data collection | Data collection have been performed by distinct TABULA partners in according to national needs. |
| E | Data quality | Data acquisition and transformation is susceptible to faults or data non availability and its directly dependent to the determination of the thermal envelope area and the conditioned floor area. |
| F | Data query | TABULA WebTool supports query: all fields of the database can be queried. |
| G | Connection with other databases and other uses | YES. It follows up on the previous EU projects DATAMINE (2006–2008). |

3.4.2. REQUEST2ACTION

The Request2Action project (presented in a nutshell in Table 7) has been carried out to stimulate uptake and investment in retrofit by ensuring easy access to accurate, trustworthy data about EPCs, bringing together market actors, households, suppliers, and policy makers through a one-stop shop model represented by the "Retrofit Action Hubs". Within this framework, five hubs have been developed within the topic of energy renovation for residential housing in the following five countries: 1. Belgium, 2. Greece, 3. Italy, 4. Portugal, and 5. the UK. The hubs were set to drive action on EPCs and they present useful and aggregated data on EPCs with the functions of market tracking data. The main idea was to create trusted meeting platforms, bringing together demand and supply of the retrofit market. The REQUEST2ACTION analysis results are presented in Table 8.

**Table 7.** REQUEST2ACTION project in a nutshell.

| Title Project | Acronym | Duration | Database Name and Link |
|---|---|---|---|
| Removing barriers to low carbon retrofit by improving access to data and insight of the benefits to key market actors | REQUEST2ACTION | 04/2014 – 08/2017 | • Belgium's Retrofit Action Hub http://genk.zetjewoningopdekaart.be/ • Greece's Retrofit Action Hub http://www.energyhubforall.eu/ • Italy's Retrofit Action Hub http://www.portale4e.it/ • Portugal's Retrofit Action Hub Portal CasA+ • Scotland's (UK) Retrofit Action Hub https://localhomesportal.est.org.uk/ |

**Table 8.** REQUEST2ACTION analysis by criteria defined in Table 1.

| Code | Criteria | Description |
|---|---|---|
| A | Organization of the data | The database collects information regarding the status of the building stock, the real energy consumption, the financing mechanisms for renovation measures, the lists of supply chain companies, etc. the information collected in the EPC databases represent a good starting point for hub population. |
| B | Organization of the database | The Hubs structure has been outlined in function of the type of information available and organized in a funnel way. |
| C | Organization responsible for the database | All REQUEST2ACTION partners are responsible for the collected data respectively at their own national level. |
| D | Data collection | Data collection have been performed by REQUEST2ACTION partners following national/regional regulations. |
| E | Data quality | Data quality check is performed by the consortium of the project. |
| F | Data query | All the Hubs are basically online tool based on GIS and in some cases, they allow users to filter the data visualization. For example, the Italian hub foresees three typologies of analysis: <br> – GIS DIPENDE (Standard web-GIS) <br> – Mapping tool <br> – Excel DIPENDE_table/queries for focus analysis |
| G | Connection with other databases and other uses | YES, with country regional and national register of EPC or system plant. For example, at the Italian level the SIAPE database collects all the EPC issued at the regional level. |

### 3.4.3. ExcEED

The ExcEED H2020 project (presented in a nutshell in Table 9) responds to the need for transparency and comparability of building energy performance calculations. The ExcEED platform gathers, categorizes, visualizes, and benchmarks multiple building data coming from building monitoring systems, projects, other EU and not-EU databases, as well as from indoor environmental quality (IEQ) surveys. The ExcEED platform collects measured quantitative (i.e., primary energy consumption, total heated floor area, etc.) and qualitative building data (i.e., occupant comfort), then uses key performance indicators to benchmark the energy efficiency of the building. ExcEED analysis results are presented in Table 10.

**Table 9.** ExcEED project in a nutshell.

| Title Project | Acronym | Duration | Database Name and Link |
|---|---|---|---|
| European Energy Efficient building district Database | ExcEED | 09/2016 – 09/2019 | enbuibench platform http://www. exceedproject.eu/register-to-the-platform/ |

**Table 10.** ExcEED analysis by criteria defined in Table 1.

| Code | Criteria | Description |
|------|----------|-------------|
| A | Organization of the data | ExcEED allows the user to upload, visualize and analyze building monitoring data through a suitable platform and two other tools: geo clustering and benchmarking tool and IEQ survey. |
| B | Organization of the database | ExcEED database is organized in 27 Key Performance Indicators (KPIs). The KPIs are organized in six categories: renewable energy, environment, technology, indoor environmental quality, electric energy, thermal energy. There are two main databases, the first one in which all the data coming from buildings are collected and used for the visualization of KPIs and the second one is mainly used to collect post-processing data used to cluster and benchmark the building. |
| C | Organization responsible for the database | The ExcEED team is made up of experts from five European organizations. The company Wattics is responsible for first database mentioned before whereas Eurac for the second one. |
| D | Data collection | The ability to integrate data streams and sets from a variety of data collection systems, files and repositories is a key element in the ExcEED platform's success. Exceed allows the user to upload, visualize and analyze building monitoring data through a suitable platform and two other tools: geo clustering and benchmarking tool and IEQ survey. To upload data the user has to follow the following instructions: 1. Definition of the organization; 2. Definition of the building; 3. Filling building metadata; 4. List of meters and sensors (i.e., temperature, humidity, energy consumption, etc.). |
| E | Data quality | All the data comes from the buildings monitoring system. There is no information about the calibration of sensors and they accuracy of each system. The tool only checks if the data uploaded are single (one value for one date) and in the right format. |
| F | Data query | Databases are not open-source, it is not possible to do query by external users. It is possible to see only aggregated and anonymous data through the geocluster tool. The rough data of each building can be visualized only by the owner. |
| G | Connection with other databases and other uses | Yes, as defined before it is possible to share the Eurac reseach database, but always checking the requirements of the GDPR. |

## 4. Overview of Building Renovation Passport Initiatives

The European policies are aimed toward significantly increasing of both the renovation rate, and the depth of energy savings in the renovation process. With the directive 2002/91/EC [9] of the European parliament and council, the EU introduces as compulsory the energy performance certificates (EPC) concerning the evaluation of the energy performance of buildings. Its primary aim was energy saving assigning energy classes to individual buildings, which inform a potential purchaser or tenant about the energy quality and consumption.

The current energy performance certificates (EPCs) provide a mere snapshot of a building's performance at a given time, lacking coherent recommendations about planned steps to bring the building to nearly zero energy building standard in the future. Building owners need in fact easily available and reliable information to drive investment decisions. The revised EPBD 2018/844 [13], among other measure that aims to accelerate the rate of building renovation towards more energy efficient systems and strengthen the energy performance of new buildings, making them smarter, introduces an optional document called a Building Renovation Passport (BRP) that is complementary to the energy performance certificates (EPC), in order to provide a long-term, step-by-step renovation roadmap for a specific building based on quality criteria, following an energy audit, and outlining relevant measures and renovations that could improve the energy performance.

In recent decades, the idea to introduce a building passport (BP) with the aim to improve the quality and quantity of communication and data between the different figures involved on the renovation process has been one of the main been discussed for decades with the objective to provide information to a potential purchaser, investors, renter, or user of the building. Today, there is not yet a common definition for BP. According to the BPIE definition [14,15], BP can be identified as a certification which provides building characteristics and technological data by collection of different documents (professionals technical reports, official declarations, system plant manuals and registers, etc.). The ALDREN project aims to explore the existing concepts and initiatives by including a detailed analysis of data accessibility and availability. This section contributes to the body of knowledge in three ways: (1) it provides an overview on BP definition from first initiatives in the EU; (2) it presents an updated evaluation and comparison of some BRP experiences developed in some European member states (Belgium, France, Germany and Denmark), selected for their advanced phase of development; (3) it points the main known barriers and the lesson learned within the review initiatives in order to provide suggestions for the standardization of BRP across the EU.

In this subsection, the state of the art [16–24] of the most developed initiatives on BRP currently ongoing in EU has been realized and summarized in Table 11 considering: (a) the general description of the building passport and (b) the respective structure.

**Table 11.** Comparison on BP initiatives across the EU.

| Passport | General Description | Structure |
|---|---|---|
| France: Passeport Efficacité Energétique | Passeperot éfficacité énergétique has been introduced by the shift project and later developed and experimented by the association Expérience P2E. The project targets about 16 million private houses in France and it aims at the massification of deep energy renovation. The French BP, considering the building's current state, identifies standard measures for renovation. | The BP has an integrated dashboard that shows the level of renovation for different building elements. Different sections show also architectural and technical characteristic of the building component. The renovation actions that compose the renovation roadmap are selected according to 36 possible actions selected by a large simulation study. |
| Germany: Sanierungsfahrplan | The Individueller Sanieringsfahrplan (iSFP) has been developed as user friendly and reliable tool considering long and short-term actions for a single-family house and multi-family house. The method is based on two site visits and on proper discussion between the owner and the auditor. It provides, through a face-to-face approach, tailored measures, starting from a standardized format. It works with the "best possible" principle, taking into account the opinion and needs of the owners. | The Sanierungsfahrplan is composed of several sheets of condensed information. The first two sections describe the motivation of the building energy renovation and the actual status of the building. The roadmap, the third section, is composed by a page overview collecting all the action planned and a detailed documentation with the presentation of different actions. |
| Belgium Flemish Region: Woningpas | Woningpas has been developed for single family house. However, it could fit also for non-residential buildings adjusting some parts of the structure. The logbook contains the all the building data collected by an on-site audit, and the official documentation (plans, invoices, permits, tax) of renovation works. The main goal is to provide a tailored renovation plan complementary to the EPC guiding as well as the owner to a higher level of awareness of the renovation strategies to apply and the operation mode of the house. | The Woningpas is intended as a digital file accessible to owners or third parties after authorization. The logbook is divided in 12 sections and it is useful to track the changes happened in the lifespan of the building. The energy module contains information regarding EP and the potential of the renovation actions. The renovation roadmap is shown in detail in a dedicated section and it contains detailed information regarding the singular action, the costs, and the energy reduction. |
| Denmark: Better Home | BetterHome is a Danish initiative initiated in 2014 and it has been developed for residential building renovation, but it can be extended to commercial buildings. The success is due to the owner centric business model and the relation instituted by user and the installers. The overall model contains 5 main phases: 1, contact the expert; 2, describe your ideas of renovation discussing target and budget available; 3, check of the building; 4, formulation of the renovation proposals; 5, propose a step-by-step renovation plan with indications of the expected benefits for each step. | The approach does not offer a long-term roadmap, but a series of tailored main renovations actions. BetterHome, through a digital platform, minimizes the extra work of the auditors with a clear step between the first contact with the owner and the finalization of the work. The auditor, using a check list, makes an overview of the building filling in the online application to calculate the energy saving potential and to extract the renovation proposal. |
| Finnish Building Passport | The FIGBC's Building Passport aims to be an accessible, visual tool that collects and presents the key futures of the buildings comparing their performances over the time. | Finnish Building Passport contains two sections: the "birth certificate" and the "health certificate" of a building. The first presents the key performance indicators specified during the design process. The "health certificate" contains the real performance of a building, update annually according to the real performances. |

## 5. Data Model Structure: Inputs, Outputs and Data Flow of the ALDREN BRP

After the identification of the dataset available on existing building stock for non-residential typology (more info available also in Annex a) and an EPC and other voluntary certification scheme available on the market and products and building components, the research work focuses on the definition of a data model to capitalize all those building information.

The data–information–knowledge–wisdom hierarchy (DIKW), referred to variously as the "Information Hierarchy" or the "Knowledge Pyramid" is one of the fundamental, widely recognized and taken-for-granted models in the information and knowledge literatures. The hierarchy is used to contextualize data, information, knowledge, and sometimes wisdom, with respect to one another and to identify and describe the processes involved in the transformation of an entity at a lower level in the hierarchy (e.g., data) to an entity at a higher level in the hierarchy (e.g., information).

The DIKW model or DIKW pyramid is an often-used method, with roots in knowledge management, to explain the ways we move from data (the D) to information (I), knowledge (K), and wisdom (W) with a component of actions and decisions. It is a model to look at various ways of extracting insights and value from all sorts of data: big data, small data, smart data, etc. In the literature, one the most known DIKW hierarchy definition is Ackoff's [25], which clearly explains that each of the higher types in the hierarchy includes the categories that fall below it. The DIKW model is quite linear and expresses a logical consequence of steps and stages with information being a contextualized 'progression' of data as it gains meaning. In Figure 1, the traditional DIKW model has been interpreted and adapted to describe the ALDREN approach.

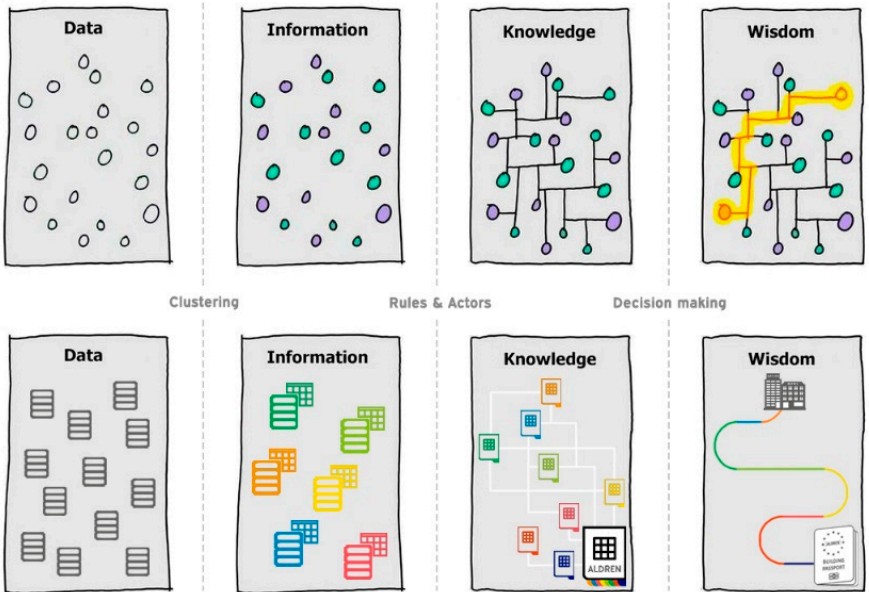

**Figure 1.** The DKIW model adopted for the ALDREN approach data model.

The D-step foresees the data collection within existing databases (i.e., European Building Stock, product databases, European projects outcomes databases, etc.) with the purpose of identifying the main data layers (such as technologies, standards, policies, buildings components, market trends, etc.) necessary for the definition of the ALDREN BRP data model. The action to reach the following I-step is data clustering: information is defined as data that is given a context or data that have been organized into a structure [26]. The fundamental principle is the refining process applied to the data, which then they evolve into information and they possess a meaningful purpose or value. Within the ALDREN approach, the clustering action has been conducted referring to the six tasks of the WP2 for the consolidation and adaptation of a European Voluntary Certification Scheme (EVCS) based on a common language.

The K-step is reached adopting rules and actors into all information clustered in the previous steps. For the ALDREN approach, this corresponds to a selection of overall indicators per topic from the raw data of each respective database. The ALDREN methodology, within this step, comprehends also an additional added value, all the overall indicators per topic are collected and linked into a unique storage: the digital version of the ALDREN BRP.

The fourth step is arguably the most elusive of all these four elements and concepts. Wisdom is the highest level of abstraction, with vision foresight and the ability to see beyond the horizon. The Awad and Ghaziri's definition of wisdom [27] is well suited to the W-step of the ALDREN approach, providing a vision foresight with the ability to communicate with core indicators—derived from the raw data of the first step—the path to follow per a specific non-residential building, the so called ALDREN renovation roadmap, and describing clearly "when, where and how renovate" buildings to

reach the NZEB target (Figure 2). Looking at the DKIW through the eyes of the ALDREN approach, it means providing concrete replies to a series of questions related to the building renovation process leads to concrete decisions and actions, increasing confidence in the investors/decision makers/owners. This is the added value of the ALDREN data model, because without actions, there is little sense in gathering, capturing, understanding, leveraging, storing, and even talking about data, information, and knowledge.

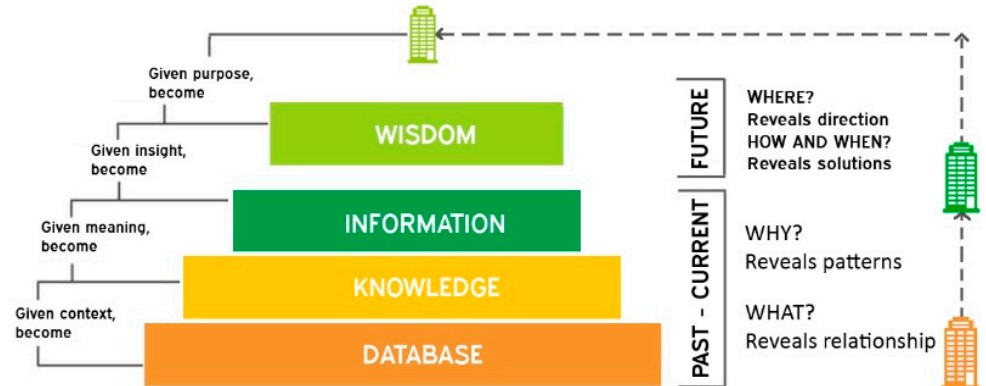

**Figure 2.** DIKW through the eyes of the ALDREN approach: the level of information and actors.

The decision to adopt the knowledge management process is because it enables actors to improve the quality of the management's decision-making by ensuring that reliable and secure information and data are available throughout all steps of the ALDREN approach, but also looking at the bottom of the pyramid as the Renovation Action—considering action as business and customers' outcomes—it means creating building value in an informed way.

The data model structure definition was the first and most fundamental step to reach the scope of Task 2.6: rendering all the collected data and results in a building renovation passport. As stated in the conclusion of the overview on the current existing initiatives on BRP, there is still not a clear a unique definition of BP, but the more used in the literature, defined by BPIE in 2016 [15], and in use also within the IBroad project [28] for the development of their outcome, foreseen the BP structure composed by two main elements: (1) a data repository, the so-called logbook and (2) the renovation roadmap. Considering this point, the basic element of the structure, the ALDREN BRP for non-residential buildings—in particular for offices and hotels—is composed of those two elements defined respectively as ALDREN BuildLog and ALDREN RenoMap.

The ALDREN BuildLog is characterized by a different level of information (LoI) to facilitate the data flow along the process and the data comprehension according to the user's expertise or needs. This concept has been represented in Figure 3 through a transposition of the traditional Russian masterpiece: the "matryoshka". Each doll represents a dataset level, and this infographic underlines the importance that no one indicator will be lost, nesting one inside the other; however, from the bigger (all raw data) to the smallest (core indicators) matryoshka, a selection from the previous LoI will be performed. The four dolls correspond to the LoI that structure the ALDREN approach and to which the different users could have access according to their needs and expertise. The big doll corresponds to the higher LoI a sort of "data lake" which contains all the information needed along the process.

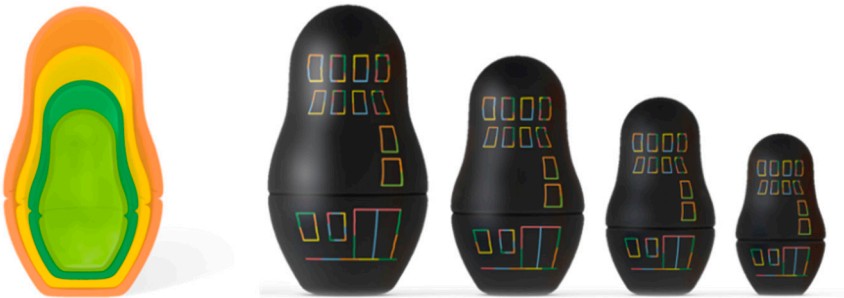

**Figure 3.** Graphical representation of the ALDREN BuildLog Lol with different data granularity.

Referring to the traditional DKIW method and starting from the first step, the ALDREN database contains all six distinct databases developed within each single task of the WP2 project: energy rating and target (Task 2.2), energy verification (Task 2.3), comfort and well-being (Task 2.4), cost, value and risk (Task 2.5), building picture (Task 2.6.2), and documentation and BIM (Task 2.6.3). Each database contains dedicated indicators, and methodology according to their respective protocols, but at the same time they are joined into a unique one, the ALDREN database, through a common language (Task 2.6.1) and some typical characteristics of a DB (i.e., database format, access, system connection, data upload, etc.) fundamental to make this repository a useful instrument.

## 6. The ALDREN BRP for Non-Residential Buildings

The directive 2018/844/EU introduced the voluntary tool of the Building Renovation Passport (BRP) in the framework of the Long Term Renovation Strategies (LTRS, Article 19a) as a possible support tool for the EPC with the main goal to provide and to identify renovation strategies in term of improving energy and comfort for the users. The ALDREN BRP focuses only on non-residential typologies in particular offices and hotels. Four are the main take-away points identified by the review of the state of the art on current BRP initiatives and on which, the ALDREN approach has been based on [29].

1. The ALDREN BRP structure is based on modules which covered different topics (energy, comfort, and costs) and provide building data (technical and non-technical) which could be updated through the time in order to have a non-static tool.
2. Each module and its respective protocol has been developed with a common language to reach different target groups of the renovation chain and to facilitate communication between them.
3. The indicators and parameters for each ALDREN BRP module has been selected referring to an agreed and valuable source (i.e., BSO, EPC, VCS, EU standards) in order to facilitate the compliancy of the methods to the already existing procedures.
4. The ALDREN BRP will contain both—the actual (as it is) and the future (after renovation)—pictures of the building with the aim to become a non-static repository for the entire life of the building, complementary to the mandatory EPC.

In this framework and keeping in mind the above listed key points, the ALDREN Building Renovation Passport has been developed to become a coherent element in a common EU solution: an instrument complementary to the EPC (ALDREN EVC), that can stimulate cost-effective renovations in the form of long-term, step-by-step renovation roadmap for a specific building based on quality criteria and following an energy audit (ALDREN BuildLog) and outlining relevant measures and renovation actions RenoMap, as suggested by Directive 2018/844/EU Art 19a.

The ALDREN BRP is composed by the two called elements: the ALDREN BuildLog and the ALDREN RenoMap, both composed by respective modules to be completed along the ALDREN approach. Figure 4 represents the data flow of the ALDREN approach that lead to the ALDREN BRP creation.

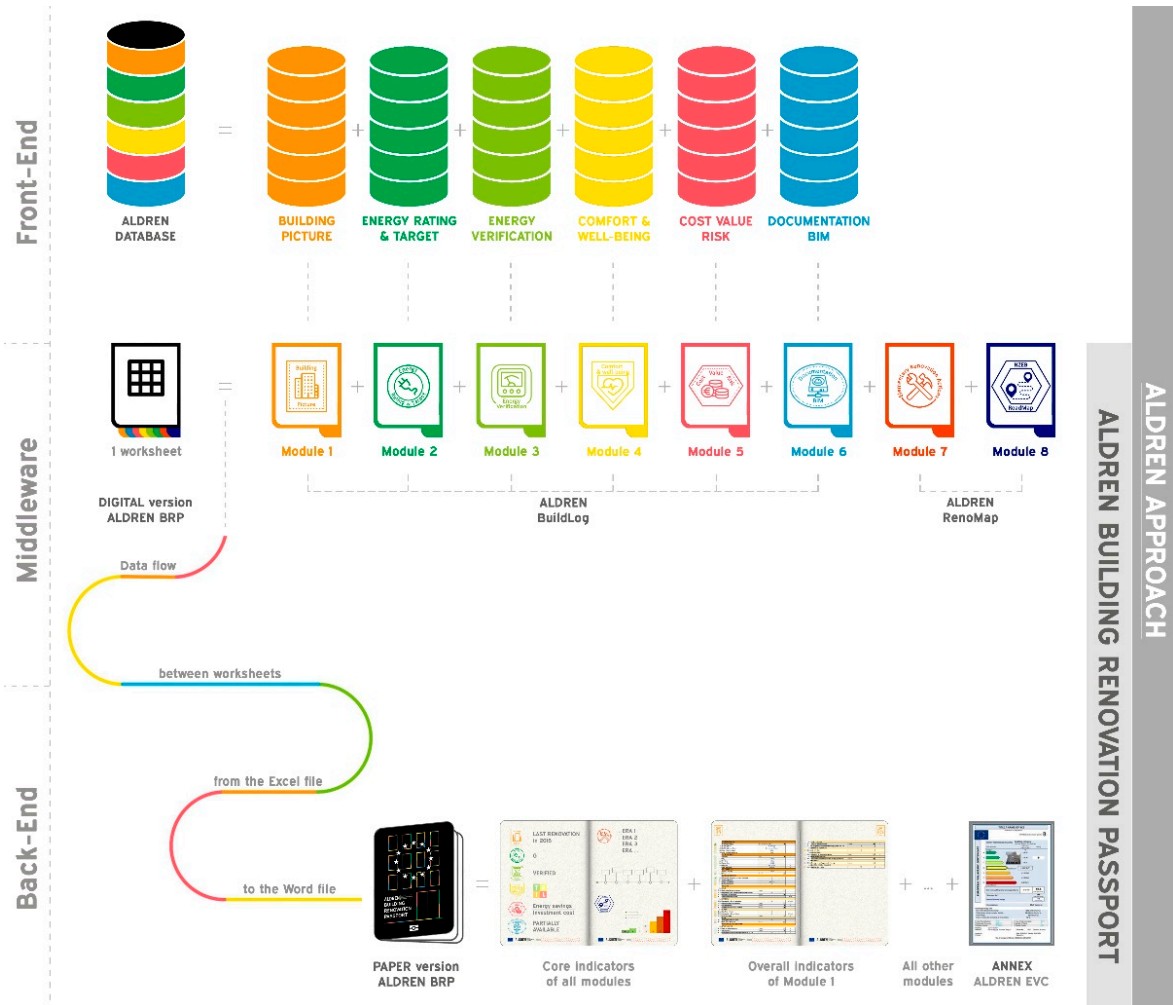

**Figure 4.** ALDREN approach scheme: a drafted data flow path for the ALDREN BRP creation.

The whole ALDREN approach foresees three main phases that could be defined with a similarity to the computer science—back-end, middleware, and front-end. The whole amount of raw data collected at the beginning of the process are joined into the ALDREN database and this corresponds to the higher LoI (back-end). Then continuing the path, in the middleware a first selection of data is performed to structure the whole spreadsheets of the ALDREN BRP digital version. The lowest LoI (front-end) is represented by the paper version of the ALDREN BRP. The possibility of having different versions of the BRP derived from the purpose to avoid the creation of a static instrument and to permit the comprehension of the data to different target groups in function of their needs and expertise.

The experts who that will be trained to apply the ALDREN approach will start the application of the approach using the excel file, having access to all the indicators collected since the databases, applying the respective protocols to manually insert the overall indicators into the excel worksheets. At the current project development, the final passage foresees the possibility of having both a paper and digital version. This user-friendly format has been chosen to increase the accessibility and feasibility of maintaining the data updates over time. Further options (i.e., implementation of the ALDREN BRP as a data layer into an open geo-mapping tool or the BSO) are currently under investigation.

The data flow of the ALDREN process and the preparation along the path of the ALDREN BRP has been represented in the graphic of Figure 5 as a time line.

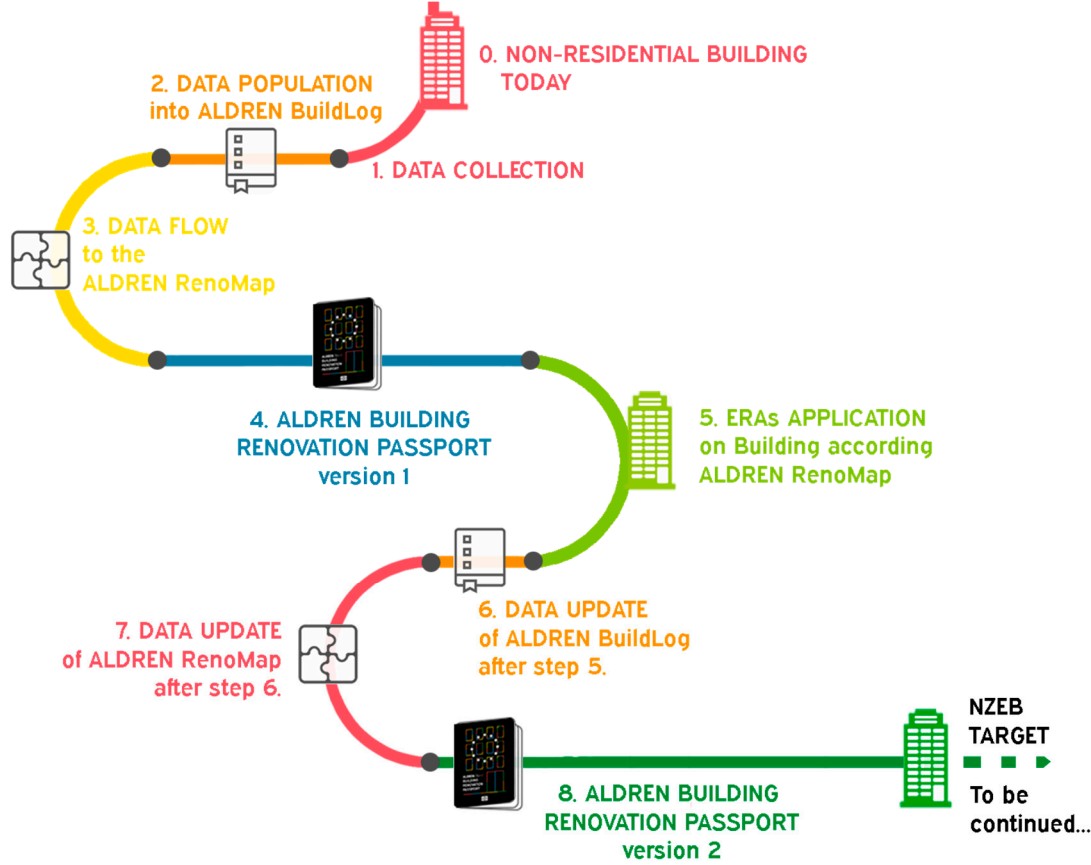

**Figure 5.** Graphical example of the ALDREN approach application and BRP updates through the time.

The beginning of the process represents the up to date conditions of the non-residential building (step 0) and it is followed by the data collection (step 1). Once collected, all the data will be populated into the ALDREN BuildLog (step 2), which is automatically connected with the RenoMap (step 3) sharing some parameters and indicators as input-output between the different sheets of the ALDREN BRP (step 4). At this point, the owner or investor could apply the ERAs individuated within the RenoMap (step 5) and once the renovation actions has been applied, the ALDREN BRP data has to be updated both in ALDREN BuildLog (step 6) and also in the ALDREN RenoMap (step 7) and consequently a version 2 of the ALDREN BRP will be generated (step 8). The process will last through the time with the final target of NZEB for the building.

## 7. Conclusions

The BRP has been structured as a supporting tool for MSs to achieve higher and deeper renovation rates considering environmental, comfort, and economic performance all linked to specific renovation processes. The ALDREN Building Renovation Passport has the role of showing clearly to the owner and/or investor a clear picture of the current state of the building (ALDREN BuildLog), providing precise technical/economic information of future steps (ALDREN RenoMap) towards low energy consumption and high IAQ (Indoor air quality). The common language, on which the ALDREN BRP is based, allows comparing different buildings and projects over the lifetime and also gives a solid basis to communicate between owners, investors, maintenance people, and tenants, and to set targets and milestones based on the life time of the building and to follow subsequent targets.

Currently the ALDREN BRP and the respective protocols, on which the ALDREN method is based, are under testing on different pilot buildings (both hotels and office buildings) around Europe. Future works will present the results of this application in order to evaluate and validate the proposed structure of the Building Renovation Passport for non-residential buildings.

**Author Contributions:** M.M.S. and G.S. conducted and lead the activities for the state of the art on the available knowledge and they performed the critical review on the current BP initiatives. M.M.S., G.S. and M.R. developed the ALDREN BRP data structure with the support of all the project partners. All authors have read and agreed to the published version of the manuscript.

**Funding:** This research was funded by European Union H2020 Work Programme, grant number 754159.

**Acknowledgments:** The work presented in this paper is part of the results obtained within the ALDREN project. The entire ALDREN project team should be acknowledged for the fruitful discussions and support. For more information on the project and partners, www.aldren.eu/.

**Conflicts of Interest:** The authors declare no conflict of interest.

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
