# Peer review of "Overview of the Available Knowledge for the Data Model Definition of a Building Renovation Passport for Non-Residential Buildings: The ALDREN Project Experience"

_sustainability, doi:10.3390/su12020642_

Round 1

Reviewer 1 Report

In this paper a study of an overview on available knowledge for the data model definition of a building renovation passport for non-residential buildings, the ALDREN project experience, is made.

Before being published, I suggest some improvements.

In the abstract I suggest the introduction of more main conclusions.

The authors should add the contribution of this stud to the state-of-the-art. What is the knowledge gap and how is this study new?

The figures can be improved.

The figure presented in the conclusion, should be moved to the results and should be discussed in the discussion area.

In the references more works from journals should be added or replaced.

In the final of the paper, the future works should be added.

Reviewer 2 Report

The manuscript does not discuss a new or an innovative subject and for this reason its scientific soundness in limited. On the other hand, it performs an interesting literature review-like description on EC databases related to the energy efficiency of buildings and the structure of an existing project and these information have a strong potential to attract readers and citations.

The text is well-written with a small exception of the last paragraph of section 1.1 that fits better to the introduction since section 1.1 is supposed to describe the concept of the ALDREN project and not the structure or the aims of the manuscript. In addition, a quick review on small typos should be done to correct minors typos like the open parenthesis in line 157 etc.

Round 2

Reviewer 1 Report

In this paper a study of an overview on available knowledge for the data model definition of a building renovation passport for non-residential buildings, the ALDREN project experience, is made.

In the actual version, in general, all suggestions given by the reviewer was implemented.